# Oblivious Sketching-based Central Path Method for Solving Linear Programming Problems

## Abstract

In this work, we propose a sketching-based central path method for solving linear programmings, whose running time matches the state of art results Cohen et al. (2019b); Lee et al. (2019). Our method opens up the iterations of the central path method and deploys an "iterate and sketch" approach towards the problem by introducing a new coordinate-wise embedding technique, which may be of independent interest. Compare to previous methods, the work Cohen et al. (2019b) enjoys feasibility while being non-oblivious, and Lee et al. (2019) is oblivious but infeasible, and relies on *dense* sketching matrices such as subsampled randomized Hadamard/Fourier transform matrices. Our method enjoys the benefits of being both oblivious and feasible, and can use *sparse* sketching matrix Nelson & Nguyên (2013) to speed up the online matrix-vector multiplication. Our framework for solving LP naturally generalizes to a broader class of convex optimization problems including empirical risk minimization.

## 1 Introduction

Linear programming is one of the fundamental models widely used in both theory and practice. It has been extensively applied in many fields such as economics Tintner (1955); Dorfman et al. (1987), operations research Delson & Shahidehpour (1992), compressed sensing Donoho (2006); Candes et al. (2006), medical studies Mangasarian et al. (1990; 1995), adversarial deep learning Wong & Kolter (2018); Weng et al. (2018), etc., due to its simple and intuitive structure. The problem of solving linear programmings has been studied since the 19-th century Sierksma & Zwols (2015).

Consider solving a general linear program in standard form $\min_{Ax=b,x\geq 0} c^\top x$ of size $A \in \mathbb{R}^{d\times n}$ without redundant constraints. For the generic case $d = \Omega(n)$ we considered in this paper, the state of art results take a total running time of $O^*(n^\omega + n^{2.5-\alpha/2} + n^{2+1/6})$[1] to obtain a solution of $\delta$ accuracy in current matrix multiplication time Cohen et al. (2019b); Lee et al. (2019), where $\omega$ is the exponent of matrix multiplication whose current value is roughly 2.373 Williams (2012); Le Gall (2014), and $\alpha$ is the dual exponent of matrix multiplication whose current value is 0.31 Le Gall & Urrutia (2018). The breakthrough work due to Cohen, Lee, and Song Cohen et al. (2019b) improves the long standing running time of $O^*(n^{2.5})$ since 1989 Vaidya (1989). For the current $\omega$ and $\alpha$, Cohen et al. (2019b) algorithm takes $O^*(n^{2.373})$ time.

For the current state-of-art results, the work Cohen et al. (2019b) involves a non-oblivious sampling technique, whose sampling set and size changes along the iterations. It avoids the possibilities of implementing expensive calculations in the preprocessing stage and also makes it harder to extend to other classical optimization problems. On the other hand, the work Lee et al. (2019) only maintains an infeasible update in each iteration and requires the usage of dense sketching matrices, which will ruin the potential sparsity structure of the original linear programs. Thus, a natural question to ask is:

---

[1]We use $O^*$ hides $n^{o(1)}$ and $\log^{O(1)}(1/\delta)$ factors.

*Is there an oblivious and feasible algorithm for solving linear programming programs in fast running time (i.e. current matrix multiplication time) ?*

In this work, we propose a both oblivious and feasible (per iteration)[2] method that solves linear programs in the same running time as the state of art.

The algorithm we propose is a *sketching-based* short step central path method. The classical short step method follows the central path in the interior of the feasible region. It decreases the complementarity gap uniformly by roughly a $1 - 1/\sqrt{n}$ factor in each iteration and takes $O^*(\sqrt{n})$ iterations to converge. This results in $O^*(\sqrt{n}) \times n = O^*(n^{1.5})$ coordinate updates throughout the algorithm Vaidya (1989). Compared to Cohen et al. (2019b); Lee et al. (2019), our randomized algorithm improves this amount (in Vaidya (1989)) of updates via a different approach. We only updating a $O^*(\sqrt{n})$-dimensional subspace in each iteration while keeping the same number of iterations $\sqrt{n}$ through an **O**blivious **C**oordinate-wise **E**mbedding (OCE) technique. Thus, our method updates $O^*(n)$ dimensions in total, which is nearly optimal.

The coordinate-wise embedding we introduce in this work is a distribution of matrices $R \in \mathbb{R}^{b_{\text{sketch}} \times n}$ with $b_{\text{sketch}} \ll n$ such that, for any inner product $g^\top h$ between two $n$-dimensional vector $g, h \in \mathbb{R}^n$, with "high" probability $g^\top R^\top R h$ approximates $g^\top h$ well. In the case of solving linear programmings, we approximate the calculation of matrix-vector multiplication $Ph$ in each iteration by $PR^\top Rh$ through OCE, such that the resulting random vector is close to previous one in each coordinate, i.e., $(PR^\top Rh)_i \approx (Ph)_i$ for all $i \in [n]$. Combining with lazy update and low-rank update techniques to maintain the query structure $PR^\top Rh$ for any input vector $h \in \mathbb{R}^n$, we can ensure the new random path is still close to the central path throughout the iterations. Therefore, our method decrease the average running time per iteration while keeping the same number of iterations. Furthermore, the sketching matrix $R$ in our approach can be chosen in an oblivious way since it does not depend on the algorithm updates. Compare to previous work Lee et al. (2019), our approximation form $PR^\top Rh$ also helps admit a closed form solution in each iteration for solving LP. Thus, our approach takes the advantages of being oblivious and feasible, compared to other state of art results Cohen et al. (2019b); Lee et al. (2019).

We state our main result as follows:

**Theorem 1.1** (Main result, informal). *Given a linear program $\min_{Ax=b,x\geq 0} c^\top x$ with no redundant constraints. Let $\delta_{\text{lp}}$ denotes precision. Our algorithm takes $O(n^{2.373} \log(n/\delta_{\text{lp}}))$ time to solve this LP.*

## 1.1 RELATED WORKS

**Linear programming.** Linear programmings have been studied for nearly a century. One of the first and most popular LP algorithm is the simplex algorithm Dantzig (1947). Despite it works well in practical small size problems, the simplex algorithm is known to be an exponential time algorithm in the worst case of Klee-Minty cube Klee & Minty (1972). The first polynomial time algorithm for solving LP is the ellipsoid method Khachiyan (1980) proposed by Khachiyan. Although this algorithm runs in polynomial time in theory, but in practice this algorithm runs much slower than the simplex algorithm. The interior point type of methods Karmarkar (1984) have both polynomial running time in theory and fast and stable performance in practice. In the case of $d = \Omega(n)$ considered in this work, Karmarkar's algorithm Karmarkar (1984) takes $O^*(n^{3.5})$ running time. Then it was improved to $O^*(n^3)$ in the work Renegar (1988); Vaidya (1987). In 1989, Vaidya further proposed an algorithm that takes a running of $O^*(n^{2.5})$. This result hasn't been improved until recent work due to Cohen, Lee and Song Cohen et al. (2019b).

**Sketching.** Classical sketching methodology proposed by Clarkson & Woodruff (2013) is the so-called "sketch and solve". The most standard and well-known applications are

---

[2]In each iteration, we approximate the central path by solving a linear system. Our approach constructs a randomized oblivious system equation which can be solved exactly. While previous work Cohen et al. (2019b) constructs a non-oblivious one, and Lee et al. (2019) doesn't solve the system exactly.

linear regression Clarkson & Woodruff (2013); Nelson & Nguyên (2013); Andoni et al. (2018); Clarkson et al. (2019); Song et al. (2019a) and low-rank approximation Clarkson & Woodruff (2013); Nelson & Nguyên (2013); Boutsidis & Woodruff (2014); Clarkson & Woodruff (2015b;a); Razenshteyn et al. (2016); Song et al. (2017; 2019b;c). It further generalizes to subspace Wang & Woodruff (2019); Li et al. (2020a), positive semi-definite matrices Clarkson & Woodruff (2017), total least regression Diao et al. (2019), quantile regression Li et al. (2020b), tensor regression Li et al. (2017); Diao et al. (2018), tensor decomposition Song et al. (2019d).

The sketching method we deploy in this work is called "'iterate and sketch" Song (2019). The major difference between classical "sketch and solve", and "iterating and sketch" is: the first one only applied the sketch once at very beginning to reduce the dimension of problem, while does not modify the solver; the second one opens up and modifies the solver by applying sketching techniques iteratively in each iteration. The idea of "iterate and sketch" has been applied to a number of problems, e.g. computing John Ellipsoid Cohen et al. (2019a), Newton method Pilanci & Wainwright (2016; 2017), tensor decomposition Wang et al. (2015); Song et al. (2016), training deep neural network Brand et al. (2020).

**Empirical risk minimization** Empirical risk minimization (ERM) problem is a fundamental question in statistical machine learning. Extensive literature has been devoted to study this topic Nesterov (1983); Vapnik (1992); Nesterov (1998); Polyak & Juditsky (1992); Nemirovski et al. (2009); Nesterov (2013); Vapnik (2013). First-order methods and a series of accelerated gradient descent algorithms for ERM are well-developed and studied Jin et al. (2018); Johnson & Zhang (2013); Nesterov & Stich (2017); Xiao & Zhang (2014); Allen-Zhu (2018). These rates depend polynomially on the smoothness/strong convexity of the objective in order to achieve a $\log(1/\epsilon)$ dependence on the error parameter $\epsilon$.

**Notations** For a positive integer $n$, we use $[n]$ to denote set $\{1, 2, \cdots, n\}$. For vectors $x, z \in \mathbb{R}^n$ and parameter $\epsilon \in (0, 1)$, we use $x \approx_\epsilon z$ to denote $(1-\epsilon)z_i \leq x_i \leq (1+\epsilon)z_i$, $\forall i \in [n]$. For any scalar $t$, we use $a \approx_\epsilon t$ to denote $(1 - \epsilon)t \leq a_i \leq (1 + \epsilon)t$, $\forall i \in [n]$. Given diagonal matrices $X = \mathrm{diag}(x) \in \mathbb{R}^{n \times n}$, $S = \mathrm{diag}(s) \in \mathbb{R}^{n \times n}$, we use $\frac{X}{S}$ to denote the diagonal matrix with $(\frac{X}{S})_{i,i} = x_i/s_i, \forall i \in [n]$.

## 2 Technique overview

In this section, we discuss the key ideas of our approach based on the central path method.

### 2.1 Short Step Central Path Method

Consider the following standard primal and dual problems of linear programmings:

$$\min_{Ax=b, x \geq 0} c^\top x \quad (\text{primal}) \quad \text{and} \quad \max_{A^\top y+s=c, \ x,s \geq 0} b^\top y \quad (\text{dual})$$

where $A \in \mathbb{R}^{d \times n}$ is full rank with $d = O(n)$. Then $(x, y, s)$ is an optimal solution if and only if it satisfies the following optimality conditions Vanderbei et al. (2015):

$$Ax = b, \ x \geq 0 \qquad \text{(primal feasibility)}$$
$$A^\top y + s = c, \ s \geq 0 \qquad \text{(dual feasibility)}$$
$$x_i s_i = 0 \text{ for all } i \qquad \text{(complementary slackness)}$$

The classical interior point method finds an optimal solution by following the central path in the interior of the feasible region, which is defined as the tuple $(x, y, s, t)$ that satisfies:

$$Ax = b, \ x > 0$$
$$A^\top y + s = c, \ s > 0 \qquad\qquad (1)$$
$$x_i s_i = t \text{ for all } i$$

where $t > 0$ is called the complementarity gap. It has been shown we can obtain an initialization point on the central path with $t = 1$ according to Ye et al. (1994). Then in each iteration, the classical algorithm deceases the complementarity gap uniformly from $t$ to

$\eta t$ with $\eta < 1$, and solves Eq. (1). As $t$ approaches 0, the central path will converge to an optimal solution. The short step central path method approximately solves Eq. (1) by the following linear system:

$$
\begin{aligned}
X\delta_s + S\delta_x &= \delta_\mu, \\
A\delta_x &= 0, \\
A^\top \delta_y + \delta_s &= 0,
\end{aligned}
\tag{2}
$$

where $X = \mathrm{diag}(x)$, $S = \mathrm{diag}(s)$ and we update the solution by $x = x + \delta_x$, $s = s + \delta_s$ and $y = y + \delta_y$. Denote the actual complementarity gap $\mu \in \mathbb{R}^n$ defined under Eq. (2) as $\mu_i = x_i s_i$ for $i \in [n]$. Then Eq. (2) maintains the feasibility conditions while approximately moving the gap from $\mu$ to $\mu + \delta_\mu$. As long as the actual complementarity gap $\mu$ is always close to the aiming complementarity gap $t$ during the algorithm, we can assure the actual complementarity gap $\mu$ will converge to 0 as $t$ goes to 0, which leads us to an optimal solution.

To solve the linear system (2), note when $A$ is full-rank, it has an unique solution explicitly given by:

$$
\delta_x = \frac{X}{\sqrt{XS}}(I - P)\frac{1}{\sqrt{XS}}\delta_\mu \text{ and } \delta_s = \frac{S}{\sqrt{XS}}P\frac{1}{\sqrt{XS}}\delta_\mu,
\tag{3}
$$

where $P = \sqrt{\frac{X}{S}}A^\top \left(A\frac{X}{S}A^\top\right)^{-1} A\sqrt{\frac{X}{S}}$ is an orthogonal projection matrix.

Literature Vaidya (1989) shows we can choose $\eta$ to be roughly $1 - \frac{1}{\sqrt{n}}$, and the algorithm converges in $O^*(\sqrt{n})$ iterations. Therefore, the total running time needed of solving LP by explicit solution Eq. (3) is $O^*(n^{\omega+1/2})$.

## 2.2 Sketching-based Central Path Method

In the following subsections, we discuss our approach of sketching-based central path method. In Subsection 2.3, we introduce the coordinate-wise embedding (CE) technique. We discuss the difference between CE and classical sketching techniques, such as Johnson-Lindenstrauss (JL) Lemma and subspace embedding (SE). We also discussed the results of applying common sketching matrices in CE. In Subsection 2.4, we explain why our sketching-based central path method can speed up the computation. In Subsection 2.5, we explain the reason why our sketch-based central path method is feasible and oblivious. In Subsection 2.6, we discuss the projection maintenance needed for the algorithm updates.

---

**Algorithm 1** Main algorithm (simplified)

---

1: **procedure** MAIN($A, b, c, \delta_{\mathrm{lp}}$)  $\qquad\qquad\qquad\qquad\qquad\qquad\qquad$ ▷ Theorem D.1
2: $\quad$ Modify the linear program and obtain an initial $x$ and $s$ according to Ye et al. (1994)
3: $\qquad\qquad\qquad\qquad\qquad\qquad\qquad$ ▷ Ensure the initial complementarity gap start with $x_i s_i = 1$
4: $\quad$ Initialize: sketching size $b_{\mathrm{sketch}} = O^*(\sqrt{n})$, parameters $\epsilon = O^*(1)$, projection maintenance datastructure mp
5: $\quad$ $t \leftarrow 1$  $\qquad\qquad\qquad\qquad\qquad\qquad\qquad\qquad\qquad\qquad$ ▷ Initialize the aiming gap $t$
6: $\quad$ **while** $t > \delta_{\mathrm{lp}}^2/(32n^3)$ **do**  $\qquad\qquad\qquad\qquad\qquad$ ▷ Stop once the precision is good
7: $\qquad$ $t^{\mathrm{new}} \leftarrow (1 - \frac{\epsilon}{3\sqrt{n}})t$  $\qquad\qquad$ ▷ Decrease the aiming gap by $1 - 1/\sqrt{n}$ in each iteration
8: $\qquad$ $\mu \leftarrow xs$  $\qquad\qquad\qquad\qquad\qquad\qquad\qquad\qquad\qquad$ ▷ Actual gap
9: $\qquad$ $\delta_\mu \leftarrow (\frac{t^{\mathrm{new}}}{t} - 1)xs - \frac{\epsilon}{2} \cdot t^{\mathrm{new}} \cdot \frac{\nabla\Phi_\lambda(\mu/t-1)}{\|\nabla\Phi_\lambda(\mu/t-1)\|_2}$  $\qquad$ ▷ Here $\Phi_\lambda(r) := \sum_{i=1}^n \cosh(\lambda r_i)$ is the potential function characterize the $\ell_\infty$ closeness between actual path $\widetilde{\mu}$ and actual path $\widetilde{t}$. We have $\Phi_\lambda$ in the update to help ensure $\widetilde{u} \approx_{0.1} t$.
10: $\qquad$ $(x^{\mathrm{new}}, s^{\mathrm{new}}) \leftarrow$ STOCHASTICSTEP(mp, $x, s, \delta_\mu, b_{\mathrm{sketch}}, \epsilon$)  $\qquad$ ▷ Algorithm 2
11: $\qquad$ $(x, s) \leftarrow (x^{\mathrm{new}}, s^{\mathrm{new}})$, $t \leftarrow t^{\mathrm{new}}$
12: $\quad$ **end while**
13: $\quad$ **return** an approximate solution of the original linear program according to Ye et al. (1994) .
14: **end procedure**

---

---

**Algorithm 2** Sketching-based central path step

---

1: **procedure** STOCHASTICSTEP(mp, $x, s, \delta_\mu, b, \epsilon$)
2:     $w \leftarrow \frac{x}{s}$, $\widetilde{v} \leftarrow$ mp.UPDATE($w$)                   ▷ Projection maintenance
3:     $\overline{x} \leftarrow x\sqrt{\frac{\widetilde{v}}{w}}$, $\overline{s} \leftarrow s\sqrt{\frac{w}{\widetilde{v}}}$         ▷ It guarantees that $\frac{\overline{x}}{\overline{s}} = \widetilde{v}$ and $\overline{x s} = xs$
4:     **repeat**
5:         $p_x, p_s \leftarrow$ mp.QUERY($\frac{1}{\sqrt{\overline{XS}}}\delta_\mu$)            ▷ Projection maintenance
6:         $\widetilde{\delta}_s \leftarrow \frac{\overline{S}}{\sqrt{\overline{XS}}}p_s$                 ▷ According to (4)
7:         $\widetilde{\delta}_x \leftarrow \frac{\overline{X}}{\sqrt{\overline{XS}}}p_x$                 ▷ According to (4)
8:     **until** $\|\overline{s}^{-1}\widetilde{\delta}_s\|_\infty \leq \frac{1}{100\log n}$ and $\|\overline{x}^{-1}\widetilde{\delta}_x\|_\infty \leq \frac{1}{100\log n}$
9:     **return** $(x + \widetilde{\delta}_x, s + \widetilde{\delta}_s)$
10: **end procedure**

---

| Statement | Reference | Emb. | $\ell_2$ **guarantee** | Unb. | Var. |
|---|---|---|---|---|---|
| Definition 2.1 | This paper | CE | for a fixed one | Yes | Yes |
| Definition 2.2 | Johnson & Lindenstrauss (1984) | JL | for a fixed set | No | No |
| Definition 2.3 | Sarlós (2006) | SE | for a subspace | No | No |

Table 1: Summary of the guarantees of different embeddings. The three embeddings give the $\ell_2$-norm guarantee for different number of vectors. Coordinate-wise embedding also guarantees the embedding is unbiased, and the variance is bounded. See Section G for more details. "Emb." denotes Embedding. "Unb." denotes Unbiased. "Var." denotes Variance.

## 2.3 COORDINATE-WISE EMBEDDING

To speed up the classical central path method, we introduce the coordinate-wise embedding (CE) as follows:

**Definition 2.1** (($\alpha, \beta, \delta$)-coordinate wise embedding (CE)). *Given parameters $\alpha, \beta \in \mathbb{R}$ and $\gamma \in (0, 1)$, we say a randomized matrix $R \in \mathbb{R}^{b_{\text{sketch}} \times n}$ with distribution $\Pi$ satisfies ($\alpha, \beta, \delta$)-coordinate-wise embedding property if for any fixed vector $g, h \in \mathbb{R}^n$, we have*

$$1. \quad \underset{R \sim \Pi}{\mathbf{E}}[g^\top R^\top R h] = g^\top h,$$

$$2. \quad \underset{R \sim \Pi}{\mathbf{E}}[(g^\top R^\top R h)^2] \leq (g^\top h)^2 + \frac{\alpha}{b_{\text{sketch}}}\|g\|_2^2\|h\|_2^2,$$

$$3. \quad \underset{R \sim \Pi}{\Pr}\left[|g^\top R^\top R h - g^\top h| \geq \frac{\beta}{\sqrt{b_{\text{sketch}}}}\|g\|_2\|h\|_2\right] \leq \delta.$$

We remark that the ($\alpha, \beta, \delta$)-coordinate wise embedding we proposed here is different from the conventional Johnson-Lindenstrauss Lemma Johnson & Lindenstrauss (1984) or subspace embedding Sarlós (2006) in classical literature.

The coordinate-wise embedding we proposed in Definition 2.1 only works for a fixed vector pair $g, h \in \mathbb{R}^n$. While for the Johnson-Lindenstrauss embedding stated below:

**Definition 2.2** (Johnson-Lindenstrauss embedding (JL) Johnson & Lindenstrauss (1984)). *Given $0 < \epsilon < 1$, a finite point set $X$ in $\mathbb{R}^n$ with $|X| = m$, we say a randomized matrix $R \in \mathbb{R}^{b \times n}$ satisfies Johnson-Lindenstrauss property if $(1 - \epsilon) \cdot \|g\|_2^2 \leq \|Rg\|_2^2 \leq (1 + \epsilon) \cdot \|g\|_2^2$ for all $g \in X$.*

The result of JL embedding works for a set of points in $\mathbb{R}^n$. However, subspace embedding holds for all the vectors from the subspace.

**Definition 2.3** (Subspace embedding (SE) Sarlós (2006)). *Given $0 < \epsilon < 1$, a matrix $A \in \mathbb{R}^{n \times d}$, we say a randomized matrix $R \in \mathbb{R}^{b \times n}$ satisfies $(1 \pm \epsilon)$ $\ell_2$-subspace embedding for the column space of $A$ if $\|RAx\|_2^2 = (1 \pm \epsilon)\|Ax\|_2^2$, for all $x \in \mathbb{R}^d$.*

The subspace embedding property works for all vectors from the subspace $\mathbb{R}^d$.

**Several well-known sketching matrices**  To further concretize our sketching approach, we discuss the following commonly used sketching matricesand their corresponding properties when acting as coordinate-wise embedding and solving LP.

**Random Gaussian matrix**  All entries are sampled from $\mathcal{N}(0, 1/b_{\mathrm{sketch}})$ independently.

**SRHT matrix Lu et al. (2013)**  Let $R = \sqrt{n/b_{\mathrm{sketch}}}SHD$, where $S \in \mathbb{R}^{b_{\mathrm{sketch}} \times n}$ is a random matrix whose rows are $b_{\mathrm{sketch}}$ uniform samples (without replacement) from the standard basis of $\mathbb{R}^n$, $H \in \mathbb{R}^{n \times n}$ is a normalized Walsh-Hadamard matrix, and $D \in \mathbb{R}^{n \times n}$ is a diagonal matrix whose diagonal elements are i.i.d. Rademacher random variables[3].

**AMS sketch matrix Alon et al. (1999)**  Let $R_{i,j} = h_i(j)$, where $h_1, h_2, \cdots, h_{b_{\mathrm{sketch}}}$ are $b_{\mathrm{sketch}}$ random hash functions picking from a random hash family $\mathcal{H} = \{h : [n] \to \{-\frac{1}{\sqrt{b_{\mathrm{sketch}}}}, +\frac{1}{\sqrt{b_{\mathrm{sketch}}}}\}\}$.

**Count-sketch matrix Charikar et al. (2002)**  Let $R_{h(i),i} = \sigma(i)$ for all $i \in [n]$ and other entries to zero, where $h : [n] \to [b_{\mathrm{sketch}}]$ and $\sigma : [n] \to \{-1, +1\}$ are random hash functions.

**Sparse embedding matrix Nelson & Nguyên (2013)**  Let $R_{(j-1)b_{\mathrm{sketch}}/s+h(i,j),i} = \sigma(i,j)/\sqrt{s}$ for all $(i,j) \in [n] \times [s]$ and all other entries to zero, where $h : [n] \times [s] \to [b_{\mathrm{sketch}}/s]$ and $\sigma : [n] \times [s] \to \{-1, 1\}$ are random hash functions.

**Uniform sampling matrix**  Let $R = \sqrt{n/b_{\mathrm{sketch}}}SD$, where $S \in \mathbb{R}^{b_{\mathrm{sketch}} \times n}$ is a random matrix whose rows are $b_{\mathrm{sketch}}$ uniform samples (without replacement) from the standard basis of $\mathbb{R}^n$, and $D \in \mathbb{R}^{n \times n}$ is a diagonal matrix whose diagonal elements are i.i.d. Rademacher random variables.

Considering an oblivious regime, where the size of sketching $b_{\mathrm{sketch}}$ is fixed, we have

**Lemma 2.4** (Oblivious coordinate-wise embedding properties). *For above defined sketching matrices, they are of $(\alpha, \beta, \delta)$-coordinate wise embedding as following:*

| Sketching matrix | $\alpha$ | $\beta$ | LP? (Left) | LP? (Right) |
|---|---|---|---|---|
| Random Gaussian | $O(1)$ | $O(\log^{1.5}(n/\delta))$ | Yes | Yes |
| SRHT | $O(1)$ | $O(\log^{1.5}(n/\delta))$ | Yes | Yes |
| AMS | $O(1)$ | $O(\log^{1.5}(n/\delta))$ | Yes | Yes |
| Count-sketch | $O(1)$ | $O(\sqrt{b_{\mathrm{sketch}}}\log(1/\delta)$ or $O(1/\sqrt{\delta})$ | No | No |
| Sparse embedding | $O(1)$ | $O(\sqrt{b_{\mathrm{sketch}}/s}\log^{1.5}(n/\delta)$ | No[†] | Yes[*] |
| Uniform sampling | $O(n)$ | $O(n/\sqrt{b_{\mathrm{sketch}}})$ | No | No |

Table 2: Summary for different sketching matrices. [*] A sparse embedding sketching matrix can be used in LP algorithm when it is added on the right and $s = \Omega(\log^3(n/\delta))$. [†] However when sketching on the left (in Lee et al. (2019)), additional algorithmic designs are needed to make the algorithm feasible (see Section F.1 for more discussion), and the error of the feasibility part cannot be bounded unless $s = \Omega(b_{\mathrm{sketch}})$.

**Remark 2.5.** *The approach in Cohen et al. (2019b) behaves similarly as applying uniform sampling matrix in our sketching, which doesn't work in an oblivious setting. Therefore, Cohen et al. (2019b) needs to modify the sketching size in each iteration. In general, to apply the sketching in an oblivious way, we observe that the sketching matrix should be relatively dense to better concentrate around its expectation, so that we can control the extra perturbation introduced by random sketching in solving linear programming problems.*

## 2.4 Speeding up central path method through OCE

To speed up the classical central path method, we randomize the calculation of Eq. (3) by

$$\widetilde{\delta}_x = \frac{X}{\sqrt{XS}}(I - P)R^\top R\frac{1}{\sqrt{XS}}\delta_\mu \text{ and } \widetilde{\delta}_s = \frac{S}{\sqrt{XS}}PR^\top R\frac{1}{\sqrt{XS}}\delta_\mu, \qquad (4)$$

---

[3]In this case, we require $\log n$ to be an integer.

where the random sketching matrix $R \in \mathbb{R}^{b_{\mathrm{sketch}} \times n}$ satisfies the $(\alpha, \beta, \delta)$-coordinate wise embedding property with $\alpha = O(1)$, $\beta = O(\sqrt{b_{\mathrm{sketch}}})$ and any failure probability $\delta \in (0, 1)$.

The coordinate-wise embedding properties 2.1 ensures the randomized Eq. (4) are well concentrated around original Eq. (3), which implies the new randomized path $\widetilde{\mu}$, will still be near the aiming path $t$ during the algorithm updates. Therefore, given the same decreasing rate of aiming path $t$ as before, we are able to prove $\widetilde{\mu} \approx_{0.1} t$ throughout the iterations. As they converge to zero, we are still able to obtain an optimal solution in $O^*(\sqrt{n})$ iterations.

In terms of running time, note Eq. (4) actually reduces the dimension of previous matrix-vector multiplication from $n$ to $b_{\mathrm{sketch}}$. Assume we can maintain the sketched projection matrix $PR^\top$ in an efficient manner, the calculation in Eq. (4) reduces to $(PR^\top) \cdot u$ for some vector $u \in \mathbb{R}^{b_{\mathrm{sketch}}}$, which is a multiplication between a matrix of size $n \times b_{\mathrm{sketch}}$ and a vector of size $b_{\mathrm{sketch}}$ and costs $O(nb_{\mathrm{sketch}})$ running time. Choosing $b_{\mathrm{sketch}}$ to be $O^*(\sqrt{n})$, we speed up the updates. We summarize our approach in Algorithm 1, 2. We discuss how to maintain the projection in Section 2.6.

## 2.5 Feasible central path equation via sketching

To explain the strength of our approach, we discuss the feasible and oblivious advantages of our method over past state-of-art results.

The new update Eq. (4) can be viewed as an **exact** solution of the following linear system:

$$
\begin{aligned}
X\widetilde{\delta}_s + S\widetilde{\delta}_x &= \widetilde{\delta}_\mu, \\
A\widetilde{\delta}_x &= 0, \\
A^\top \widetilde{\delta}_y + \widetilde{\delta}_s &= 0,
\end{aligned}
\tag{5}
$$

where

$$
\widetilde{\delta}_\mu = \sqrt{XS} R^\top R \frac{1}{\sqrt{XS}} \delta_\mu.
\tag{6}
$$

Therefore, our approach can also be viewed as an update of a subspace of the complementarity gaps in each iteration, instead of decreasing the complementarity gaps uniformly.

Note in each iteration, our update solves the new linear system (5) **exactly**. Compared to the state-of-art approach Lee et al. (2019) which constructs a solution that solves the linear system **inexactly**, the feasibility of our approach prevents us from the complicated analysis. Our approach is also able to use sparse embedding matrix to prevent ruining the potential sparsity structure of the original linear programs, compared to the usage of dense sketching matrices in the work Lee et al. (2019).

On the other hand, our method is oblivious since the choice of sketching matrix $R \in \mathbb{R}^{b_{\mathrm{sketch}} \times n}$ **does not depend** on the algorithm updates, which implies we can pick the sketching matrices at the preprocessing stage. While for the state-of-art approach Cohen et al. (2019b), its sampling probability **depends** on the algorithm updates and needs to be calculated on-the-fly.

## 2.6 Projection maintenance

In this section, we discuss our approach to deal with the second computational bottleneck, i.e., how to maintain the projection after sketching $PR^\top \in \mathbb{R}^{n \times b_{\mathrm{sketch}}}$ in an efficient way, where $P \in \mathbb{R}^{n \times n}$ is the orthogonal projection matrix defined in Eq. (3) and $R \in \mathbb{R}^{b_{\mathrm{sketch}} \times n}$ is a random sketching matrix with appropriate $(\alpha, \beta, \delta)$-coordinate-wise embedding property.

Let $W := \mathrm{diag}(w) \in \mathbb{R}^n$ denotes the diagonal matrix with $w_i = x_i/s_i$. Then we have $P := \sqrt{W} A^\top (AWA^\top)^{-1} A \sqrt{W} \in \mathbb{R}^{n \times n}$. Therefore, our final goal of implementing of Eq. (4) reduces back to the task of maintaining the query structure which outputs:

$$
PR^\top Rh = (PR^\top) \cdot (R \cdot h) = (PR^\top) \cdot u
\tag{7}
$$

where $u \in \mathbb{R}^{b_{\mathrm{sketch}}}$.

To achieve this, we have the similar observation as in Cohen et al. (2019b): $W$ doesn't vary much between two iterations under the sketching approach, which is shown in the following lemma:

**Lemma 2.6** (Change of $W$). *Let $0 < \epsilon < 1/(40000 \log n)$. Let $w_i$ and $w_i^{\text{new}}$ denote the value $x_i/s_i$ in two consecutive iterations, then we have*

$$\sum_{i=1}^{n} (\mathbf{E}[\ln w_i^{\text{new}}] - \ln w_i)^2 \leq 64\epsilon^2, \quad \sum_{i=1}^{n} (\mathbf{Var}[\ln w_i^{\text{new}}])^2 \leq 1000\epsilon^2.$$

Above observation motivates us to take the benefit of lazy update if $w_i$ only has little changes since we only need to maintain the projection approximately. We discuss two extreme scenarios to illustrate the core ideas: 1) $w$ changes uniformly across all coordinates and 2) $w$ only changes in few coordinates.

In the first case, we use the idea of lazy update. Lemma 2.6 implies the changes of $w$ between two iterations are roughly $w_i^{\text{new}} \approx (1 \pm \frac{1}{\sqrt{n}})w_i$. Therefore, $w_i$'s will only vary by more than a constant and possibly ruin our previous $\ell_\infty$ closeness after $O^*(\sqrt{n})$ number of iterations. In this case, we only need to update the matrix $PR^\top$ once every $O^*(\sqrt{n})$ iterations, while being "lazy" in any other time. Since the algorithm finishes in $O^*(\sqrt{n})$ iterations, it means we only need to update matrix $PR^\top$ $O^*(1)$ many times, whose total running time is $O^*(n^\omega)$.

In the second case, we use the idea of low-rank update. Instead of updating $PR^\top$ in each iteration, we directly compute $PR^\top u$ using the Woodbury matrix identity. Since $w$ only changes in few coordinates, we only need to focus on computing the inverse of small matrix instead of the original $n \times n$ matrix. And computing $PR^\top u$ instead of $PR^\top$ fastens the computation because we only need to do matrix-vector multiplication. As a result, we can output $PR^\top u$ in $O^*(nb_{\text{sketch}})$ time. Recall we choose $b_{\text{sketch}} = O^*(n^{1/2})$. Therefore, the running time for this case is $O^*(n^2)$, which is also within our budget.

For general cases, we combine above techniques together and have the following theorem:

**Theorem 2.7** (Projection maintenance). *Given a number $a \in (0, \alpha)$[4] and sketching matrices $R \in \mathbb{R}^{n^b \times n}$ with $b \in [0, 1]$. We can approximately maintain the projection by*

1. UPDATE($w$): *Output a vector $\widetilde{v}$ such that for all $i$,*
$$(1 - \epsilon_{\text{mp}})\widetilde{v}_i \leq w_i \leq (1 + \epsilon_{\text{mp}})\widetilde{v}_i.$$

2. QUERY($h$): *Output $\sqrt{\widetilde{V}}A^\top (A\widetilde{V}A^\top)^{-1}A\sqrt{\widetilde{V}}(R^\top)_{*,l}R_{l,*}h$ for the $\widetilde{v}$ outputted by the last call to UPDATE.*

*The data structure takes $n^2 d^{\omega - 2}$ time to initialize, each call of QUERY($h$) takes time $O^*(n^{1+b} + n^{1+a})$, and the amortized expected time per call of UPDATE($w$) is $O^*(n^{\omega - 1/2} + n^{2-a/2})$.*

Note our approach maintains the projection at point $\widetilde{V} = \overline{X}/\overline{S}$ instead of $W = X/S$, where $\overline{x}_i = x_i \sqrt{\widetilde{v}_i/w_i} \approx_{0.1} x_i$ and $\overline{s}_i = s_i \sqrt{w_i/\widetilde{v}_i} \approx_{0.1} s_i$. Equivalently speaking, instead of solving Eq. (5), we are solving

$$\overline{X}\delta_s + \overline{S}\widetilde{\delta}_x = \widetilde{\delta}_\mu,$$
$$A\widetilde{\delta}_x = 0, \tag{8}$$
$$A^\top \widetilde{\delta}_y + \widetilde{\delta}_s = 0,$$

where $\widetilde{\delta}_\mu = \sqrt{\overline{X}\overline{S}}R^\top R\frac{1}{\sqrt{\overline{X}\overline{S}}}\delta_\mu$ as shown in Algorithm 2. And the final running time of our algorithm can be bounded bt $O^*(n^\omega)$.

**Corollary 2.8** (Extension to ERM). *Our approach for solving LP naturally generalizes to other convex optimization problems of the following form Lee et al. (2019), including empirical risk minimization: $\min_x \sum_i f_i(A_i x + b_i)$, where $f_i$ is convex function on $\mathbb{R}^{n_i}$ with $n = \sum_i n_i$. Our algorithm output the solution in time $O^*(n^{2.373} \log(n/\delta))$, where $\delta$ is the precision parameter.*

---

[4]$\alpha$ is the dual exponent of matrix multiplication, whose current value is roughly 0.31 Le Gall & Urrutia (2018).

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
