# OpenReview forum: "Oblivious Sketching-based Central Path Method for Solving Linear Programming Problems"
_ICLR.cc/2021/Conference — Reject_

### Official Review · AnonReviewer1 · 2020-10-28
**theoretical result, unclear implications**

**Rating:** 4
**Confidence:** 3

**Review:**

This paper improves robust central path based methods for linear programming. It's able to keep intermediate matrices oblivious, and solutions feasible. It builds upon a line of recent work that solves linear programs in time proportional to the cost of inverting a single matrix.

In my opinion, these improved methods are of mostly theoretical interest: if one assume n^{3} time matrix multiplication, the rank-1 update schemes from the original Karmarkar paper already obtain an n^{3} runtime. Such gains also hold for more intermediate values of matrix multiplication (e.g. Strassen fast matrix multiplication has bene implemented, so 2.8 is a reasonable assumption from practical perspectives), but I think the 2-batch update scheme of Vaidya is already sufficient as well.

Therefore, I'm concerned whether this paper is suitable for this venue. There are no experimental evaluations of the algorithm, and the theoretical gains are too reliant on the matrix multiplication exponent being < 2.5.

---

> ### Author Response · Authors · 2020-11-25
> **Thank you for the comments!**
>
> We thank the reviewer for the comments! We address the concerns as follows:
>
> Firstly, we point out that our contributions are mainly theoretical. In theory, the value of \omega is only about 2.37, and the first result showing \omega less than 2.38 is in 1990, by Don Coppersmith and Shmuel Winograd [1], which is 30 years ago.
>
> Secondly, we notice that in the sequential implementation, the practical exponent of matrix multiplication is about 2.77 [2]. And in current Python implementations, we actually allow parallel implementation, which can potentially decrease the exponent of matrix multiplication to be below 2.5 with careful engineering.
>
> Lastly, we remark that the ICLR conference does accept many theoretical works each year. And one of the previous works [Lee et al. COLT 19] was also published at a learning conference.
>
> [1] https://en.wikipedia.org/wiki/Coppersmith%E2%80%93Winograd_algorithm
> [2] Pan, Victor Y. "Fast feasible and unfeasible matrix multiplication." arXiv preprint arXiv:1804.04102 (2018).

---

### Official Review · AnonReviewer3 · 2020-10-29
**This paper analyzes a randomized linear program solver that employs oblivious sketching and maintains feasibility. However, the requirement of feasibility and oblivious sketching is not well motivated.**

**Rating:** 5
**Confidence:** 3

**Review:**

Summary:  The authors propose a randomized linear program solver that applies sketching to the central path calculations. The resulting algorithm maintains feasibility and employs oblivious sketching matrices as opposed to other randomized solvers. The algorithm is relatively simpler to describe. However, the requirement of feasibility and oblivious sketching is not well motivated.


Major comments/questions
1. The difference between oblivious and non-oblivious sketches.
The advantage of the oblivious sketching is not well explained and motivated. Is there an algorithmic advantage of the oblivious sketching methodology? Oblivious sketching has clear advantages in streaming data settings, however it's not clear what would be the benefit in an iterative algorithm with an obvious sketch. Given that the computational complexity is the same as leverage score sampling based non-oblivious sketching based methods, the authors need to provide a stronger motivation.

2. Maintaining feasibility.
It's not clear why exact feasibility is important in this class of randomized algorithms. Maintaining approximate feasibility should be possible to obtain relatively easily in the state of the methods. Please correct me if I'm wrong.

3. Sparsity.
The authors mention destroying the sparsity pattern as a shortcoming. Does the proposed method leverage input sparsity and sparsity pattern? Is it possible to quantify this, e.g., in terms of the number of nonzero entries?


Minor comments/questions

1. Left and right sketch.
Can you clarify what is the algorithmic consequence of the left/right sketching property in Table 2.

2. Coordinate-wise embedding
It looks like coordinate wise embedding property is a standard property that is satisfied for almost all known randomized JL embeddings, as opposed to a new construction. Is there a known JL construction that doesn't satisfy the coordinate-wise embedding property?

3. In corollary 2.8, what are the dimensions of A_i ?

Minor comment
1. Some small typos
page 1 footnote 1. Missing word or typo in 'We use O* hides;
page 2 'linear programmings'. I believe 'linear programs' is more idiomatic.

---

> ### Author Response · Authors · 2020-11-19
> **Thank you for the comments!**
>
> We thank the reviewer for the comments! We address these questions as follows:
>
> > The difference between oblivious and non-oblivious sketches. The advantage of the oblivious sketching is not well explained and motivated. Is there an algorithmic advantage of the oblivious sketching methodology? Oblivious sketching has clear advantages in streaming data settings, however it's not clear what would be the benefit in an iterative algorithm with an obvious sketch. Given that the computational complexity is the same as leverage score sampling based non-oblivious sketching based methods, the authors need to provide a stronger motivation.
>
> We point out that compared to the previous non-oblivious method, our oblivious method is more likely to apply to the setting where we want to solve LP problems in the streaming model. To have a taste of what the benefits will be, we refer the reviewer to see the work of maximum weight matching [1].
>
> > Maintaining feasibility. It's not clear why exact feasibility is important in this class of randomized algorithms. Maintaining approximate feasibility should be possible to obtain relatively easily in the state of the methods. Please correct me if I'm wrong.
>
> We admit that coming up with methods with approximate feasibility is easier. But we also need to point out the corresponding analysis for infeasible methods will be unnecessarily complicated. Examples of such complicated analysis for infeasible methods include [2][3].
>
> > Sparsity. The authors mention destroying the sparsity pattern as a shortcoming. Does the proposed method leverage input sparsity and sparsity pattern? Is it possible to quantify this, e.g., in terms of the number of nonzero entries?
>
> We point out that in Table 2 --- Sparse embedding matrix, we show that our approach can use sketching matrices with polylog(n) sparsity, while the previous approach in the work of Lee et al. (2019), which sketches from the left, can not. We emphasize that this type of contribution is highly non-trivial. We refer to the following work discussing the importance of using sparse sketching matrices: [4][5][6][7].
>
> > Left and right sketch. Can you clarify what is the algorithmic consequence of the left/right sketching property in Table 2.
>
> We point out that there is a huge difference in the algorithmic low-level side. Once applying sketching from the left-hand side, we will meet the infeasibility issue, which requires extra effort to take care of converting infeasibility back to feasibility. This will make the algorithm side way more complicated and requires extra proof. On the other hand, however, sketching from the right-hand side will simplify things a lot, due to the fact that we do not need to deal with the infeasibility anymore.
>
> > Coordinate-wise embedding It looks like coordinate wise embedding property is a standard property that is satisfied for almost all known randomized JL embeddings, as opposed to a new construction. Is there a known JL construction that doesn't satisfy the coordinate-wise embedding property?
>
> Commonly seen unbiased JL embeddings with bounded variance are always coordinate-wise embedding.
>
> > In corollary 2.8, what are the dimensions of A_i ?
>
> The dimension of A_i is n_i by d.
>
> [1] Liu, S. Cliff, Zhao Song, and Hengjie Zhang. "Breaking the $ n $-Pass Barrier: A Streaming Algorithm for Maximum Weight Bipartite Matching." arXiv preprint arXiv:2009.06106 (2020).
> [2] van den Brand, Jan, et al. "Solving tall dense linear programs in nearly linear time." Proceedings of the 52nd Annual ACM SIGACT Symposium on Theory of Computing. 2020.
> [3] Brand, Jan van den, et al. "Bipartite Matching in Nearly-linear Time on Moderately Dense Graphs." arXiv preprint arXiv:2009.01802 (2020).
> [4] Clarkson, Kenneth L., and David P. Woodruff. "Low-rank approximation and regression in input sparsity time." Journal of the ACM (JACM) 63.6 (2017): 1-45.
> [5] Nelson, Jelani, and Huy L. Nguyên. "OSNAP: Faster numerical linear algebra algorithms via sparser subspace embeddings." 2013 ieee 54th annual symposium on foundations of computer science. IEEE, 2013.
> [6] Cohen, Michael B., T. S. Jayram, and Jelani Nelson. "Simple analyses of the sparse Johnson-Lindenstrauss transform." 1st Symposium on Simplicity in Algorithms (SOSA 2018). Schloss Dagstuhl-Leibniz-Zentrum fuer Informatik, 2018.
> [7] Kane, Daniel M., and Jelani Nelson. "Sparser johnson-lindenstrauss transforms." Journal of the ACM (JACM) 61.1 (2014): 1-23.

---

### Official Review · AnonReviewer2 · 2020-10-29
**Oblivious sketching based interior point method matching state-of-the-art LPs**

**Rating:** 4
**Confidence:** 4

**Review:**

SUMMARY: The paper is based on the works of Lee et al and Cohen et al. Building upon these works, the paper comes up with an interior point method that matches the state-of-the-art in LPs. The paper's contribution is in a new type of sketching used inside the interior point method, that demonstrates some advantages over those of Lee et al and Cohen et al. Notable among these advantages are (1) the ability to preserve sparsity of an LP, and (2) exact solutions to the system of linear equations obtained from optimality conditions.

REVIEW: I don't think the paper is ready for publication yet, since it needs significant polish and strengthening. My reasons are stated below.

1. My biggest issue with the paper is that the motivation doesn't seem strong enough to me. If, despite being able to use exact solvers in each step of the IPM, the paper is able to merely match the run time of Lee et al and Cohen et al, then what is the advantage of doing the exact solves? I think the paper could be greatly strengthened if they are able to apply this technique to *improving* run times of some problem classes, as opposed to just matching state-of-the-art on one problem class.

2. While one could argue that using exact solvers and having feasibility in each step is theoretically interesting, I personally don't think it's interesting enough to merit publication at ICLR; of course, this is a subjective matter of taste, so one could argue otherwise as well. However, I think if this were to be of interest to the broader theoretical CS audience (beyond the handful of researchers that have worked on this problem in the recent past), then the paper's scope would need to be wider. Can the authors try to make more changes to these methods and *still* match the state-of-the-art? With the current modifications, to me, the paper doesn't seem sufficiently different from previous works to merit publication.

3. The sketching techniques shown are not different enough. I have the same comment about it as 1. and 2.

4. I feel the writing also needs to be significantly improved. Leaving aside error of language and grammar, what I think needs a vast improvement is the story-telling aspect of the paper. There needs to be greater effort put into telling *why* characteristic X of algorithm A is "bad" or "good", *why* some question is interesting, etc. I think, at its current state, the paper can be read and comprehended by a very small handful of people (the ones whose works are cited), and that, to me, is really not enough.

5. Finally, I was disappointed that a paper marketed as so heavily theoretical, there is not even a proof sketch shown of the high level technique or any of the technical contributions. I shouldn't have to go to the appendix to see these, and the authors should put some effort into putting some math into the main body. I believe that highlighting novel proof techniques in the main body would also go a long way into convincing the reader(s) (such as myself) of novelty of the paper's contributions.

---

> ### Author Response · Authors · 2020-11-19
> **Thank you for the comments!**
>
> We thank the reviewer for the valuable comments! We address these concerns as follows.
>
> > My biggest issue with the paper is that the motivation doesn't seem strong enough to me. If, despite being able to use exact solvers in each step of the IPM, the paper is able to merely match the run time of Lee et al and Cohen et al, then what is the advantage of doing the exact solves? I think the paper could be greatly strengthened if they are able to apply this technique to improving run times of some problem classes, as opposed to just matching state-of-the-art on one problem class.
>
> We sincerely believe matching instead of improving upon the state of art results is not an issue here, since our methods and techniques are different from the previous approaches. And these new techniques and methods give different insights into the problem and have a potential impact in the field. Also, the result of solving linear programs in running time n^\omega actually matches the running time of solving a linear system Ax=b, indicating it is already optimal in some sense. Just for the record, the work of Lee et al is published in COLT 2019 and the work [1] is published in SODA 2020. These results also just match the work of Cohen et al, without giving further improvement.
>
> > The sketching techniques shown are not different enough. I have the same comment about it as 1. and 2.
>
> We emphasize that our sketching technique ---- coordinate-wise embedding (CE), is quite different from other classical sketching techniques such as JL embedding and subspace embedding (SE). Especially, our proposed algorithms and analysis build upon this special sketching technique. To see the difference between our sketching technique and classical sketching techniques, we present concrete and detailed comparison between them in Table 1, where the main differences are from three perspectives: 1) different \ell_2-norm guarantee: CE argues for a fixed point, JL argues for a finite set of fixed points, SE argues for a subspace; 2) CE guarantees an unbiased embedding, while others do not; 3) CE guarantees the embedding to have bounded variance while others do not.
>
> > Finally, I was disappointed that a paper marketed as so heavily theoretical, there is not even a proof sketch shown of the high level technique or any of the technical contributions. I shouldn't have to go to the appendix to see these, and the authors should put some effort into putting some math into the main body. I believe that highlighting novel proof techniques in the main body would also go a long way into convincing the reader(s) (such as myself) of novelty of the paper's contributions.
>
> We point out that we present the proof sketch in Section 2. Our proof sketch contains 3 parts:
> Part 1) Our whole method is based on the short step central method (Section 2.1), where we aim to implement Eq.(3) efficiently (O(n^{\omega - ½})) running time in average) while keeping the same number of iterations (O(n^{½})). Note there are two computational bottlenecks in implementing Eq.(3): 1) matrix-vector multiplication requires n^2 running time and 2) matrix inverse in calculating the projection matrix P requires n^\omega running time. Both of these calculations exceed our “budget”.
> Part 2) To deal with the first bottleneck, we propose our novel sketching technique --- coordinate-wise embedding (see Section 2.3) to speed up the matrix-vector multiplication. With appropriate sketching matrices, we can randomize the update of Eq.(3) by Eq.(4) without ruining the central path, i.e., the update of Eq.(4) is well concentrated around Eq.(3) (see Section 2.4). Since the sketching decreases the dimension of the matrix-vector multiplication, we resolve this bottleneck.
> Part 3) To deal with the second bottleneck, we propose a special update mechanism to maintain the projection (after sketching). We analyze two extreme cases to illustrate the idea (see Section 2.6): 1) w_i’s change uniformly: we propose the “lazy update” technique, which only updates when the changes are large enough (once every O*(\sqrt{n}) iterations); 2) w only changes in O(1) number of coordinates: we propose the “low-rank update” technique, which is based upon the Woodbury matrix identity. We only update the coordinates that are changed to reduce the calculation. Combining both techniques, we are able to maintain the projection in O(n^{\omega-½}) time per iteration on average, which resolves the second bottleneck.
> To conclude, we present the proof sketch along with the novel techniques such as coordinate-wise embedding, lazy update, low-rank update, etc. in Section 2. And our overall analysis is a sophisticated combination of these involved techniques.
>
> [1] van den Brand, Jan. "A deterministic linear program solver in current matrix multiplication time." Proceedings of the Fourteenth Annual ACM-SIAM Symposium on Discrete Algorithms. Society for Industrial and Applied Mathematics, 2020.

---

> > ### Comment · AnonReviewer2 · 2020-11-20
> > **Replying to Comment**
> >
> > > since our methods and techniques are different from the previous approaches. And these new techniques and methods give different insights into the problem and have a potential impact in the field.
> >
> > I think the paper needs to be specific about this claimed potential impact. What exactly is the impact you foresee?
> >
> > > Also, the result of solving linear programs in running time n^\omega actually matches the running time of solving a linear system Ax=b, indicating it is already optimal in some sense.
> >
> > I know that $n^\omega$ is optimal for this problem, which is why I am not following the significance of your submission; since the works you mention already got the optimal result, what is the added contribution or novelty of your submission?
> >
> > > Just for the record, the work of Lee et al is published in COLT 2019 and the work [1] is published in SODA 2020. These results also just match the work of Cohen et al, without giving further improvement.
> >
> > The publications of these results is not of any relevance at all in discussing the publication of your submission.
> >
> >  I am sorry, but the proof sketch from Section 2 is essentially identical to that of Cohen, Lee, and Song. They also have sqrt(n) iterations at per iteration cost of n^{omega - 1/2}. I really don't see the novelty in this submission, I am sorry.

---

> > > ### Author Response · Authors · 2020-11-25
> > > **Thank you for your reply!**
> > >
> > > We thank the reviewer for the reply! We address these questions as follows:
> > >
> > > > I think the paper needs to be specific about this claimed potential impact. What exactly is the impact you foresee?
> > >
> > > We believe the potential impact of our new techniques and methods includes but is not limited to 1) application towards more non-convex optimization problems, e.g. training neural network models [1]; 2) application in the field of reinforcement learning, where the underlying Markov decision process (MDP) can be written as a specific linear program [2][3][4][5].
> > > [1] Brand, Jan van den, et al. "Training (Overparametrized) Neural Networks in Near-Linear Time." ITCS 2021.
> > > [2] Wang, Mengdi. "Randomized linear programming solves the Markov decision problem in nearly linear (sometimes sublinear) time." Mathematics of Operations Research 45.2 (2020): 517-546.
> > > [3] Sidford, Aaron, et al. "Solving discounted stochastic two-player games with near-optimal time and sample complexity." International Conference on Artificial Intelligence and Statistics. PMLR, 2020.
> > > [4] Sidford, Aaron, et al. "Near-optimal time and sample complexities for solving Markov decision processes with a generative model." Advances in Neural Information Processing Systems. 2018.
> > > [5] Sidford, Aaron, et al. "Variance reduced value iteration and faster algorithms for solving Markov decision processes." Proceedings of the Twenty-Ninth Annual ACM-SIAM Symposium on Discrete Algorithms. Society for Industrial and Applied Mathematics, 2018.
> > >
> > > > I know that nω is optimal for this problem, which is why I am not following the significance of your submission; since the works you mention already got the optimal result, what is the added contribution or novelty of your submission?
> > >
> > > We propose different methods and techniques to solve the problem, which has potential impacts on future work and other problems. The differences are specified in the next reply.
> > >
> > > > With the current modifications, to me, the paper doesn't seem sufficiently different from previous works to merit publication.
> > >
> > > > but the proof sketch from Section 2 is essentially identical to that of Cohen, Lee, and Song. They also have sqrt(n) iterations at per iteration cost of n^{omega - 1/2}.
> > >
> > > We thank the reviewer for pointing this out and we address these concerns together! We claim that our approach is actually very different from the previous state of the art works.
> > > Specifically, 1) comparison to the work of Cohen, Lee, and Song (CLS): Despite both our methods and their methods are built upon the short step central path method (which as the reviewer has pointed out, have sqrt{n} iterations at per iteration cost of n^{omega - 1/2}”), the oblivious sketching method we use to accelerate the central path iterations are totally different from the non-oblivious sampling approach proposed by CLS. As pointed out in Remark 2.5, Section 2.5 in the main text, and Section F in the supplementary material, the sampling technique proposed by CLS behaves similarly to a specific uniform sketching in our framework. However, such uniform sketching is incapable to implement in an oblivious way as shown in Table 2 --- Uniform sampling. Therefore, such a sampling approach requires the change of sampling distribution along with the iterations, while our proposed sketching method works in an oblivious way with generated random sketching matrices before the iterations of the algorithm.
> > > And 2) comparison to the work of Lee, Song, and Zhang (LSZ): As pointed out in Section 2.5 in the main text and detailedly discussed in Section F in the supplementary material, the major difference between our approach and the approach of LSZ is we sample from different directions: the work of LSZ tries to randomize the short step central path method by applying sketching matrices from the left-hand-side, while ours applies sketching from the right-hand-side. The resulting differences include 1) our updates after sketching can be naturally interpreted as the solution of a randomized short-step equation, while theirs can not (see discussion in Section F in the supplementary material, and 2) the choices of sketching matrices that can be used in our approach are much more than the choices of sketching matrices in LSZ (e.g., see Table 2, we can use sparse embedding matrices while LSZ can not).

---

### Official Review · AnonReviewer4 · 2020-10-30
**Nice result with theoretical guarantees**

**Rating:** 7
**Confidence:** 3

**Review:**

The paper studies the problem of solving LP and more generally convex programming via sketching based approaches. In particular, the running time of proposed algorithm in this paper matches the running time of the best known algorithms [Cohen et al(19b) and Lee et al(19)]. However, this paper provides some further useful properties: 1) oblivious sketching, 2) sparse sketching which can be of interest in many applications. The paper has provided theoretical guarantees of their proposed algorithm. The problem that is studied in this paper is very important and the sketching based approach is also very practical. The paper would benefit from an empirical comparison of the proposed algorithm with the existing methods for solving LPs.

Also, one way of motivating the new approach is the use of sparse sketching methods. Recently for several tasks in numerical linear algebra it has been shown that NN can learn efficient sparse sketching matrices which outperform the standard "randomized" sketching constructions such as CountSketch.

However, my only concern is whether ICLR is the right venue for this purely theoretical paper. While I am supportive of the paper I can understand the potential objections regarding lack of empirical evaluations.

Minor Comments:
- fix quotation marks (e.g., the "iterate and sketch" in the abstract.)
- page 2: We only updating -> We only update
- page 2: our method decrease -> our method decreases
- page 2: sketching methodology proposed by Clarkson and Woodruff'13. I would instead say Sarlos'06 and Clarkson and Woodruff'13
- page 3:  generalizes to subspace -> subspace embedding
- page 4: Literature Vaidya (1989) shows -> The work of ... shows that
- page 4: ... potential function characterize -> charcterizing
- page 5: rephrase the sentence before Def 2.2
- page 6: matricesand -> matrices and
- page 7: state-of-art -> state of the art (fix all instances)
- page 7: The new update ->The new update rule
- page 7: solve the linear system inexactly -> solve the linear system approximately?
- page 7: the feasibility of our approach prevents us from the complicated analysis -> maybe need to rephrase it?
- page 7: implementing of Eq. (4) reduces back -> implementing Eq. (4) reduces
- page 8: we only need to do matrix-vector multiplications. How many such multiplications are needed? O(1)? specify it.
- page 8: fasten -> speed up?

=====POST-REBUTTAL COMMENTS========
As I mentioned earlier in my review, I like the result and I feel it could be interest of the TCS community. However, as correctly raised by other reviewers, ICLR may not be the right venue for this paper and also it would be beneficial if authors improve the presentation of their result and the motivation of their work further.

---

> ### Author Response · Authors · 2020-11-19
> **Thank you for the comments!**
>
> We thank the reviewer for the valuable comments! We have fixed all the typos and grammar issues pointed out in the updated version.

---

### Decision · Program_Chairs · 2021-01-07
**Final Decision**

**Decision:**

Reject

**Comment:**

The reviewers found it hard to understand the motivation of using both oblivious sketching and maintaining feasibility throughout the course of the algorithm, given that the ultimate running times matched those of existing work. Because there wasn't a concrete improvement over prior work, the worry is what the impact of the paper would ultimately be. There was also a concern with novelty, similarity to the work of Cohen, Lee, and Song, and a reliance on fast matrix multiplication exponents. The paper could also benefit from an improved presentation.